

# Retrieval of snow layer and melt pond properties on Arctic sea ice from airborne imaging spectrometer observations

Sophie Rosenburg[1*], Charlotte Lange[1*], Evelyn Jäkel[1], Michael Schäfer[1], André Ehrlich[1], and Manfred Wendisch[1]

[1]Leipzig Institute for Meteorology (LIM), Leipzig University, Stephanstr. 3, 04103 Leipzig, Germany
[*]These authors contributed equally to this work.

**Correspondence:** Sophie Rosenburg (sophie.rosenburg@uni-leipzig.de)

**Abstract.** A melting snow layer on Arctic sea ice, as a composition of ice, liquid water, and air, supplies meltwater that may trigger the formation of melt ponds. As a result, surface reflection properties are altered during the melting season and thereby may change the surface energy budget. To study these processes, sea ice surface reflection properties were derived from airborne measurements using imaging spectrometers. The data were collected over the closed and marginal Arctic sea ice zone north of Svalbard in May/June 2017. A retrieval approach based on different absorption indices of pure ice and liquid water in the near-infrared spectral range was applied to the campaign data. The technique enables to retrieve the spatial distribution of the liquid water fraction of a snow layer and the effective radius of snow grains. For observations from three research flights liquid water fractions between $8.7\%$ and $15.6\%$ and snow grain sizes between $115\,\mu m$ and $378\,\mu m$ were derived. In addition, the melt pond depth was retrieved based on an existing approach that isolates the dependence of a melt pond reflectance spectrum on the pond depth by eliminating the reflection contribution of the pond ice bottom. The application of the approach to several case studies revealed a high variability of melt pond depth with maximum depths of $0.33\,m$. The results were discussed considering uncertainties arising from the reflectance measurements, the setup of radiative transfer simulations, and the retrieval method itself. Overall, the presented retrieval methods show the potential and the limitations of airborne measurements with imaging spectrometers to map the transition phase of the Arctic sea ice surface, examining the snow layer composition and melt pond depth.

## 1 Introduction

Compared to the globe, the Arctic experiences an enhanced warming, which is referred to as Arctic amplification (Serreze and Francis, 2006; Serreze and Barry, 2011). The snow-ice-surface-albedo feedback is one of the most important mechanisms driving Arctic amplification (Curry et al., 1995; Hall, 2004; Pithan and Mauritsen, 2014; Wendisch et al., 2023). The Arctic sea ice albedo depends on wavelength, solar zenith angle, snow grain size, and shape as well as snow layer morphology, impurities, and liquid water fraction. Therefore, the sea ice albedo is strongly altered by melting processes (Warren, 1982; Kokhanovsky and Zege, 2004; Dozier et al., 2009; Gardner and Sharp, 2010).



Following the snow metamorphism, the deposited snow grains become more spherical and larger, leading to a decrease of surface albedo (Warren, 1982; Colbeck, 1983; Gubler, 1985). During the summer months, the initially dry and cold snow layer covering the sea ice surface is beginning to melt and thereby undergoing three melting stages: moistening, ripening, and runoff (Dingman, 2015). Meltwater accumulates in the initially air-filled interstices between the snow grains leading to a further surface albedo decrease. In this stage, the melting snow layer is composed of a mixture of ice, liquid water, and air, as schematically illustrated in Fig. 1 (I. Snow melting). If the maximum snow grain interstitial capacity is reached, the runoff phase begins (Dingman, 2015). Meltwater accrues in sea ice surface depressions and melt ponds form (Polashenski et al., 2012), as illustrated in Fig. 1 (II. Ponding). The meltwater volumes stored in melt ponds, depending on surface area and depth, represent a significant portion of the ice surface meltwater balance (Perovich et al., 2021). Overall, Fig. 1 demonstrates the sea ice surface transition from a melting snow layer to beginning melt pond formation in late spring and early summer, which is characterized by a distinct surface albedo decrease (Perovich and Polashenski, 2012). To observe this phase in more detail,

**Figure 1.** Schematic overview of the sea ice surface transition during the early melting season: a melting snow layer (I.) as a composition of snow grains, liquid water, and interstitial air, determining the ongoing albedo decrease. With start of the runoff phase, the melt pond formation (II.) is induced. The reflective behavior of the melt pond is described by its depth and the ice bottom albedo, indicated by a color gradient.

the snow grain size, snow layer wetness, and melt pond depth are important parameters characterizing the melting processes. Past Arctic field campaigns provided in situ surface albedo measurements over a melting snow layer and melt ponds (Perovich et al., 2002; Light et al., 2022). The retrieval of the regarded properties was already subject of several studies. For example, Jäkel et al. (2021) compared snow grain size retrieval methods based on the grain size-dependent absorption in the solar spectral range, which were applied to ground-based, airborne, and spaceborne reflectance measurements. Grain sizes below $300\,\mu\text{m}$ were retrieved for springtime snow layers on sea ice. Hannula and Pulliainen (2019) examined the snow reflectance in visible to near-infrared spectral bands as a function of wetness in a laboratory experiment. Marin et al. (2020) investigated the information on snow wetness in spaceborne radar observations.





To quantify the snow layer wetness, the liquid water fraction $f_{LW}$ is a useful measure. It is defined as the ratio of snow layer liquid water content (LWC) and total water content (TWC = ice water content + LWC), which are both given in units of $\mathrm{g\,m^{-3}}$. Therefore, the $f_{LW}$ of a snow layer can range between $0\,\%$ (dry snow) and $15\,\%$ (very wet snow) reaching a soaked state with $f_{LW} > 15\,\%$ (Fierz et al., 2009). A snow reflectance spectrum is sensitive to the snow layer wetness in the near-infrared spectral range because of different absorption characteristics of liquid water and pure ice (Warren, 1982; Kou et al., 1993). Based on the spectral dependence of local absorption minima and maxima, Green et al. (2002) retrieved snow layer liquid water fraction and snow grain size by comparing measured snow reflectance spectra with simulations for varying snow grain sizes and liquid water fractions. This approach was tested under laboratory conditions by Green et al. (2002) and validated by Donahue et al. (2022) with first field experiments.

The reflectance of melt ponds depends on the melt pond ice bottom reflectance and pond depth (Malinka et al., 2018). Based on this dependence, several approaches retrieving the pond depth were developed (Legleiter et al., 2014; Malinka et al., 2018; Lu et al., 2018). König and Oppelt (2020) derived a linear model to isolate the dependence of the pond reflectance spectrum on the pond depth. The depth of melt ponds is depending on sea ice surface topography with shallower/deeper ponds covering undeformed first-year/deformed multi-year ice and reaches at maximum $1\,\mathrm{m}$ (Untersteiner, 1961; Morassutti and LeDrew, 1996; König et al., 2020; Webster et al., 2022).

In this study, the retrievals of snow layer liquid water fraction and snow grain size as well as melt pond depth are based on reflectance measurements. The spectral reflectance $\mathcal{R}_\lambda$ is defined here as:

$$\mathcal{R}_\lambda = \frac{\pi \cdot I_\lambda^{\uparrow}}{F_\lambda^{\downarrow}}\,\mathrm{sr}\,, \tag{1}$$

with the upward spectral radiance $I_\lambda^{\uparrow}$ in units of $\mathrm{W\,m^{-2}\,nm^{-1}\,sr^{-1}}$ and the downward spectral irradiance $F^{\downarrow}$ in units of $\mathrm{W\,m^{-2}\,nm^{-1}}$. Reflectance measurements could be performed on ground-based, airborne, or spaceborne platforms. However, for observing surface features, airborne reflectance measurements have the advantage of providing data with higher spatial resolution than spaceborne sensors and a greater spatial coverage in contrast to ground-based measurements. In this study, the approaches by Green et al. (2002) and König and Oppelt (2020) are adapted and applied to airborne imaging spectrometer observations, captured in the framework of an Arctic field campaign performed in May/June 2017, for selected case studies. As a result, for the first time snow layer liquid water fraction, snow grain size, and melt pond depth are derived from airborne imaging spectrometer observations to enable a combined analysis of the snow layer and melt pond state during the early stages of the melting season. Providing a technical perspective, the present paper evaluates the potential as well as limitations of these retrieval methods and is structured as follows. The airborne measurements and the setup for snow layer radiative transfer simulations are introduced in Sect. 2. The study is further subdivided into two main parts, the retrieval of snow layer properties in Sect. 3 and the retrieval of melt pond depth in Sect. 4, which comprise the approach methodology and the results, respectively. Following a discussion of technical limitations in Sect. 5, a conclusive summary is given in Sect. 6.





## 2 Data and tools

### 2.1 Airborne measurements

Airborne observations of sea ice surface characteristics were performed during the Arctic CLoud Observations Using airborne measurements during polar Day (ACLOUD) campaign from 23 May to 26 June 2017 (Wendisch et al., 2019). The research flights covered the north-west of Svalbard (Fig. 2). The *Polar 5* aircraft of the Alfred Wegener Institute, Helmholtz Centre for Polar and Marine Research (Wesche et al., 2016) was equipped with remote sensing instruments measuring solar spectral radiation (Ehrlich et al., 2019), providing spectral surface reflectance measurements according to Eq. 1. The specifications of these instruments are summarized in Table 1 and explained in the following.

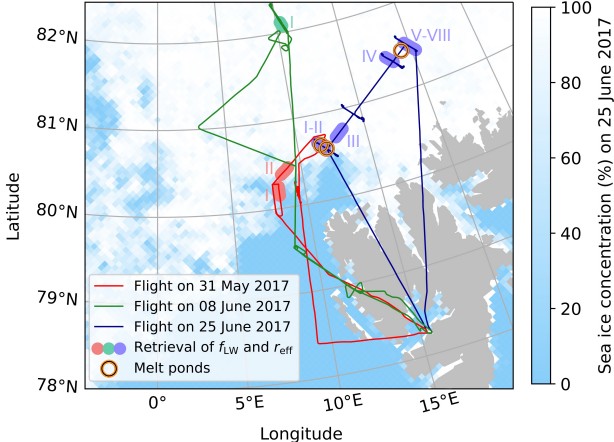

**Figure 2.** Map showing three flight tracks of the aircraft *Polar 5* during the ACLOUD campaign with highlighted and numbered segments (several overflights in case of the flight on 25 June 2017), for which the liquid water fraction $f_{LW}$ and the effective radius $r_{eff}$ were retrieved. Locations of the selected melt ponds are marked by orange open circles. In the background the AMSR2 sea ice concentration on the 25 June 2017 is shown (Spreen et al., 2008).

The Spectral Modular Airborne Radiation measurement sysTem (SMART) albedometer was installed to measure the solar spectral downward and upward irradiance with 2 Hz temporal resolution (Wendisch et al., 2001; Bierwirth et al., 2009; Ehrlich et al., 2019; Jäkel et al., 2021). For each hemisphere an optical inlet was mounted on the aircraft fuselage, connected via optical fibres to two respective spectrometers (Wendisch and Mayer, 2003). A wavelength range from 400 nm to 2150 nm is covered with a full width at half maximum (FWHM) for each spectrometer of $1 - 2$ nm and $9 - 16$ nm, respectively. The optical inlets were actively stabilized to account for the varying aircraft attitude with an accuracy of $\pm 0.2\%$ (Wendisch et al., 2001) for pitch and roll angles in a range of $\pm 4.5°$. Considered uncertainties account for the cosine correction (4 %) and sensor tilt (2.5 %). Further uncertainties include the wavelength accuracy as well as contributions from the radiometric calibration. The laboratory calibration was transferred to field conditions using a transfer calibration regularly performed during the airborne campaign





**Table 1.** Characteristics of the SMART albedometer (two spectrometers) and the imaging spectrometers AisaEagle and AisaHawk (FWHM - full width at half maximum, FOV - field of view).

|  | SMART | AisaEagle | AisaHawk |
|---|---|---|---|
| quantity (unit) | $F^{\downarrow}$ (W m$^{-2}$ nm$^{-1}$) | $I^{\uparrow}$ (W m$^{-2}$ nm$^{-1}$ sr$^{-1}$) | $I^{\uparrow}$ (W m$^{-2}$ nm$^{-1}$ sr$^{-1}$) |
| spectral range (nm) | $400-1000; 1000-2150$ | $400-990$ | $940-2500$ |
| spectral resolution (nm) | 0.8; 5.1 | 1.2 | 5.6 |
| FWHM (nm) | $1-2; 9-16$ | 1.2 | 5.6 |
| FOV (°) | 180 | 36 | 36 |
| spatial pixels | $-$ | 1024 | 384 |
| temporal resolution (Hz) | 2 | 20 | 20 |
| uncertainty (%) | $\pm 5.7$ | $\pm 3$ | $\pm 3; \pm 3.5; \pm 4$ |

(see Sect. 5.2). A total uncertainty of $\pm 5.7\,\%$ for the downward irradiance in the near-infrared spectral range was estimated by Jäkel et al. (2021).

AisaEagle and AisaHawk are across track pushbroom imaging spectrometers with a field of view (FOV) of $36°$, which is spatially divided into 1024 (AisaEagle) and 384 (AisaHawk) pixels. These instruments measure the upward radiance with $20\,\mathrm{Hz}$ temporal resolution, covering collectively a wavelength range from $400\,\mathrm{nm}$ to $2500\,\mathrm{nm}$ (Schäfer et al., 2013; Ehrlich et al.,

2019; Ruiz-Donoso et al., 2020). The radiance measurements have a spectral, temporal, and spatial dimension. An AisaEagle or AisaHawk scene is composed of a swath of pixels moving forward due to the aircraft motion. Therefore, the area covered by a single pixel is determined by the FOV and the number of spatial pixels as well as the flight altitude and aircraft speed (Schäfer et al., 2013). The calibration of the instruments was performed with a certified diffuse radiation source, whose relative uncertainty varies spectrally. For the spectral range of the AisaEagle radiance relevant for this study the calibration uncertainty

amounts to $\pm 3\,\%$ ($500-990\,\mathrm{nm}$). The radiance measured by AisaHawk is required for a wider spectral range, for which the calibration uncertainty varies between $\pm 3\,\%$ ($940-990\,\mathrm{nm}$), $\pm 3.5\,\%$ ($1000-1100\,\mathrm{nm}$), and $\pm 4\,\%$ ($1150-1700\,\mathrm{nm}$).

## 2.2    Radiative transfer simulations

In order to simulate snow reflectance spectra, the library of radiative transfer routines and programs (*libRadtran*) was used (Emde et al., 2016; Mayer et al., 2019). Applying *libRadtran* to model the radiative transfer in a dense medium such as a snow

layer requires that the far field assumption applies, which presumes that particles are at distance and, therefore, the scattering waves can be assumed to be planar. Additionally, the multiple scattering assumption needs to be valid that defines particles by their single-scattering properties and assumes no interaction between the particles takes place. Both assumptions might be violated, when increasing the cloud density more than hundredfold to represent a snow layer. The issue was addressed by Pohl et al. (2020), who showed corresponding effects can be neglected.





The optical properties of the snow layer were calculated for a gamma size distribution $n(L)$ (Emde et al., 2016) with the maximal dimension $L$, effective area $A$ and volume $V$. The size of ice particles or liquid water spheres in the snow layer is represented by the effective radius $r_{\text{eff}}$,

$$r_{\text{eff}} = \frac{3}{4} \frac{\int_{L_{\min}}^{L_{\max}} V(L) \cdot n(L) \, \mathrm{d}L}{\int_{L_{\min}}^{L_{\max}} A(L) \cdot n(L) \, \mathrm{d}L}. \tag{2}$$

For our purpose the database of optical properties available in *libRadtran* was expanded to simulate effective particle radii $r_{\text{eff}}$ larger than $25 \, \mu\text{m}$. The single scattering properties (single scattering albedo, extinction coefficient) and Legendre moments representing the scattering phase function of ice crystals with sizes up to $800 \, \mu\text{m}$ were taken from an external data base (Yang et al., 2000). The "smooth droxtal" shape was selected since it accounts for the expected rounding of ice crystals during the snow ageing process. Applied by Pohl et al. (2020), this particle shape is assumed to be an adequate choice. For liquid water

spheres the Mie-tool, provided by *libRadtran* (Wiscombe, 1980), was used to derive respective single scattering properties. The $\delta$-M-approach (Wiscombe, 1977) was applied in the simulations in order to reduce the number of Legendre moments necessary for an adequate representation of the scattering phase function. The bulk optical properties were scaled accordingly. More detailed information on the simulation setup are provided in the Appendix A.

## 3 Retrieval of snow layer properties

### 3.1 Methodology

To retrieve maps of snow layer particle size and liquid water fraction an approach by Green et al. (2002) was adapted. Their approach is based on a least square fit between measured and simulated snow layer reflectance spectra in the near-infrared spectral range, in which the local maxima of liquid water and ice absorption indices are shifted by several nanometers. Thus, this spectral range of a snow layer reflectance spectrum is characterized by the liquid water fraction and the effective radius

of snow grains. A direct derivation of $f_{\text{LW}}$ from the spectral shift of the reflectance minimum was not feasible due to its nonlinearity and sensitivity with respect to grain size and observation angle. Furthermore, the spectral resolution of the imaging spectrometers is too low to resolve the nearly sigmoidal spectral shift function. Therefore, the retrieval method by Green et al. (2002) was adapted and applied to selected measurement cases observed during ACLOUD and *libRadtran* simulations.

The selection of ACLOUD flight sections used in this study was based on certain criteria. Overall, only cloud-free conditions

were considered to reduce the required input information for radiative transfer simulations. Furthermore, flight sections with temporal stability of aircraft heading and height as well as pitch and roll angles near $0°$ were selected. Hence, eleven flight sections from flights on 31 May 2017, 08 June 2017, and 25 June 2017 were chosen. They are depicted in Fig. 2, the specific times of the selected flight sections are provided in Table 2.

For the retrieval of $r_{\text{eff}}$ and $f_{\text{LW}}$, the AisaHawk measurements ($20 \, \text{Hz}$ resolution) were smoothed to fit the SMART measure-

ments ($2 \, \text{Hz}$ resolution). This reduces the influence of small spatial structures and three-dimensional (3D) effects. Both spectral





data sets were interpolated to a common wavelength grid with a spectral resolution of $\Delta\lambda = 1\,\mathrm{nm}$. Using the upward radiance from the AisaHawk instrument and the downward irradiance data from the SMART albedometer, the spectral reflectance was calculated according to Eq. 1. In accordance to the libRadtran simulations, the reflectance spectra of the AisaHawk swath were averaged to 13 observation angles between $\alpha = -15°$ and $+15°$ in $\Delta\alpha = 2.5°$ steps to reduce the influence of local inhomo-

geneities. This resulted in a nadir pixel area of $4.4\,\mathrm{m} \times 30\,\mathrm{m}$ (across $\times$ along track) for a flight altitude of $100\,\mathrm{m}$, aircraft speed of $60\,\mathrm{m\,s^{-1}}$, and $0.5\,\mathrm{s}$ integration time.

Regarding the simulations, temporally constant conditions throughout each individual flight section were assumed and hence, the respective reflectance spectra were calculated for the averaged solar azimuth and zenith angle, aircraft height and heading. The melting snow layer was assumed to be a mix of liquid water spheres in between droxtal shaped ice particles. Donahue

et al. (2022) also applied the approach by Green et al. (2002) and showed this interstitial sphere model to be the most reliable out of three different models they tested in comparison to laboratory and field experiments.

This way a Look-up-Table (LUT) was simulated with *libRadtran* for varying observation angles, effective radii, and liquid water fractions. In the LUTs, the effective radii were varied between $r_{\mathrm{eff}} = 50\,\mu\mathrm{m}$ and $800\,\mu\mathrm{m}$ in $\Delta r_{\mathrm{eff}} = 50\,\mu\mathrm{m}$ steps, and liquid water fractions ranged from $f_{\mathrm{LW}} = 0\,\%$ to $30\,\%$ in $\Delta f_{\mathrm{LW}} = 2.5\,\%$ steps. The total water content was set to $\mathrm{TWC} =$

$100{,}000\,\mathrm{g\,m^{-3}}$ and the snow thickness to $1\,\mathrm{m}$. The albedo of the underlying surface was chosen to be zero. This is justified, because the TWC was chosen sufficiently high to assure that the reflectance is independent of the underlying surface albedo as most of the scattering takes place in the upper few centimeters of the snow layer.

In order to reduce the influence of the wavelength-dependent systematic errors in the instrumental calibration, simulated and measured reflectance spectra were normalized by the respective reflectance value at the wavelength $\lambda = 1100\,\mathrm{nm}$, where the

absorption indices of liquid water and ice are the same. Hence, there is no sensitivity to the liquid water fraction at this point, while it is maintained at all other wavelengths. In a last step, the simulated LUTs were convoluted according to the AisaHawk slit function (Ehrlich et al., 2019).

In case of the $f_{\mathrm{LW}}$-retrieval, the wavelength range between $\lambda = 982 - 1054\,\mathrm{nm}$ was chosen for the least square fit between measured and simulated reflectance spectra (Fig. 3, Part 1). It covers the reflectance minimum for pure ice at $1030\,\mathrm{nm}$ and

omits areas with strong atmospheric absorption.

In case of the $r_{\mathrm{eff}}$-retrieval, three wavelength ranges were selected for the least square fit (Fig. 3): $\lambda = 982 - 1054\,\mathrm{nm}$ (Part 1), $\lambda = 1181 - 1240\,\mathrm{nm}$ (Part 2) and $\lambda = 1294 - 1320\,\mathrm{nm}$ (Part 3). In addition, an alternative approach to retrieve $r_{\mathrm{eff}}$-maps was derived, which is based on spectral characteristics of the reflectance spectrum. All simulated spectra showed a constant reflectance in the wavelength range $\lambda = 1240 - 1295\,\mathrm{nm}$ (Fig. 4a, black lines). In this spectral range the normalized

reflectance depends on $r_{\mathrm{eff}}$, while being mostly independent of $f_{\mathrm{LW}}$. By averaging all simulated spectra for one certain $r_{\mathrm{eff}}$ over all simulated $f_{\mathrm{LW}}$, a reference curve was derived for each flight section and observation angle (Fig. 4b). It links the mean normalized reflectance over $\lambda = 1240 - 1295\,\mathrm{nm}$ to a certain effective radius. By interpolating the reference curves, a resolution of $\Delta r_{\mathrm{eff}} = 1\,\mu\mathrm{m}$ was achieved. The effective radius of the observed snow surface pixel could then be derived by comparing the mean of the measured reflectance spectrum over $\lambda = 1240 - 1295\,\mathrm{nm}$ to the reference curve, again omitting the atmospheric

absorption band in-between (Fig. 4a, red lines).

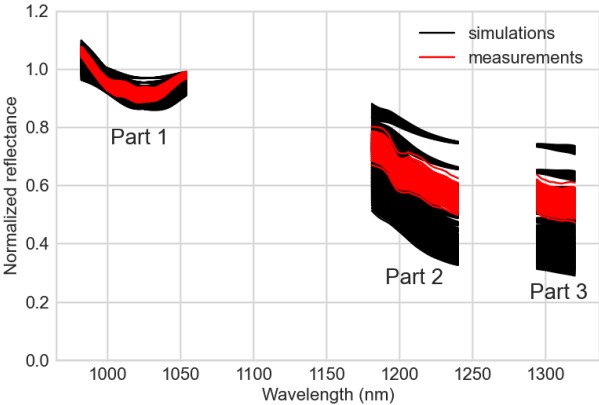

**Figure 3.** Comparison of measured (red) and simulated (black) reflectance spectra for flight section 2017/06/25 (I) for nadir measurements. Each black spectrum accounts for a certain combination of $r_{\mathrm{eff}}$ and $f_{\mathrm{LW}}$, each red spectrum for one time step of the flight section. The wavelength ranges used for the least square fit are indicated (Part 1-3). For the derivation of $f_{\mathrm{LW}}$ Part 1, for $r_{\mathrm{eff}}$ Parts 1-3 were used.

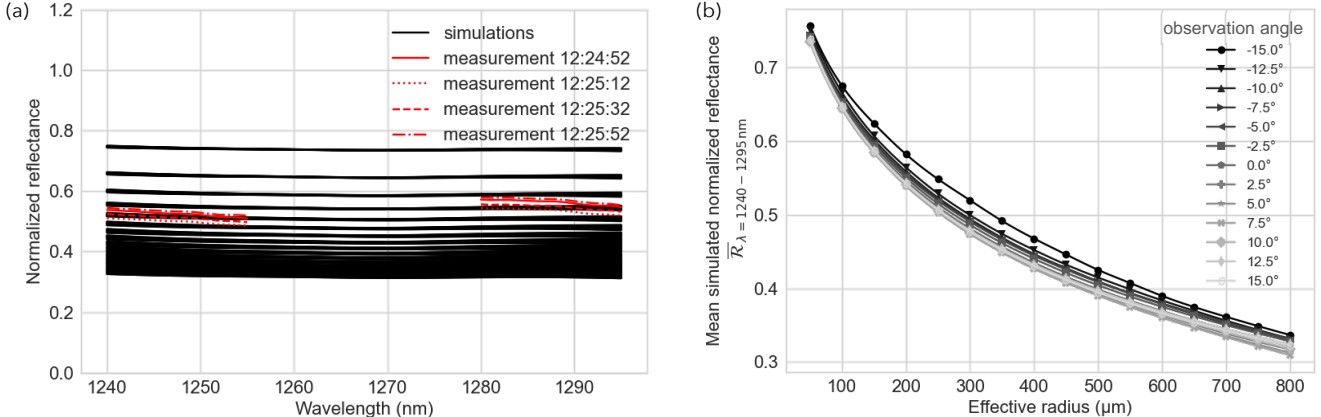

**Figure 4. (a)** The simulations (black) were averaged between $\lambda = 1240 - 1295\,\mathrm{nm}$ over all $f_{\mathrm{LW}}$ for each $r_{\mathrm{eff}}$, creating a reference curve for each observation angle shown in **(b)**. The mean of each measured spectrum (four example spectra with given time in UTC shown in red) in **(a)**, omitting an atmospheric absorption band between $\lambda = 1255 - 1280\,\mathrm{nm}$, was compared to the reference curve in **(b)** and thereby $r_{\mathrm{eff}}$ was derived. This was done for every flight section and observation angle, here exemplarily shown for flight section 2017/06/25 (I).

## 3.2 Retrieval results

We applied the retrieval methods to derive spatial maps of $r_{\mathrm{eff}}$ and $f_{\mathrm{LW}}$ for eleven selected flight sections. A statistical overview of the results is given in Table 2. Exemplarily, Fig. 5 shows the AisaEagle-RGB-composite, maps, and frequency distributions of $r_{\mathrm{eff}}$ (reference curve retrieval) and $f_{\mathrm{LW}}$ (least square retrieval) for flight section 2017/06/25 (I). The parameter maps show



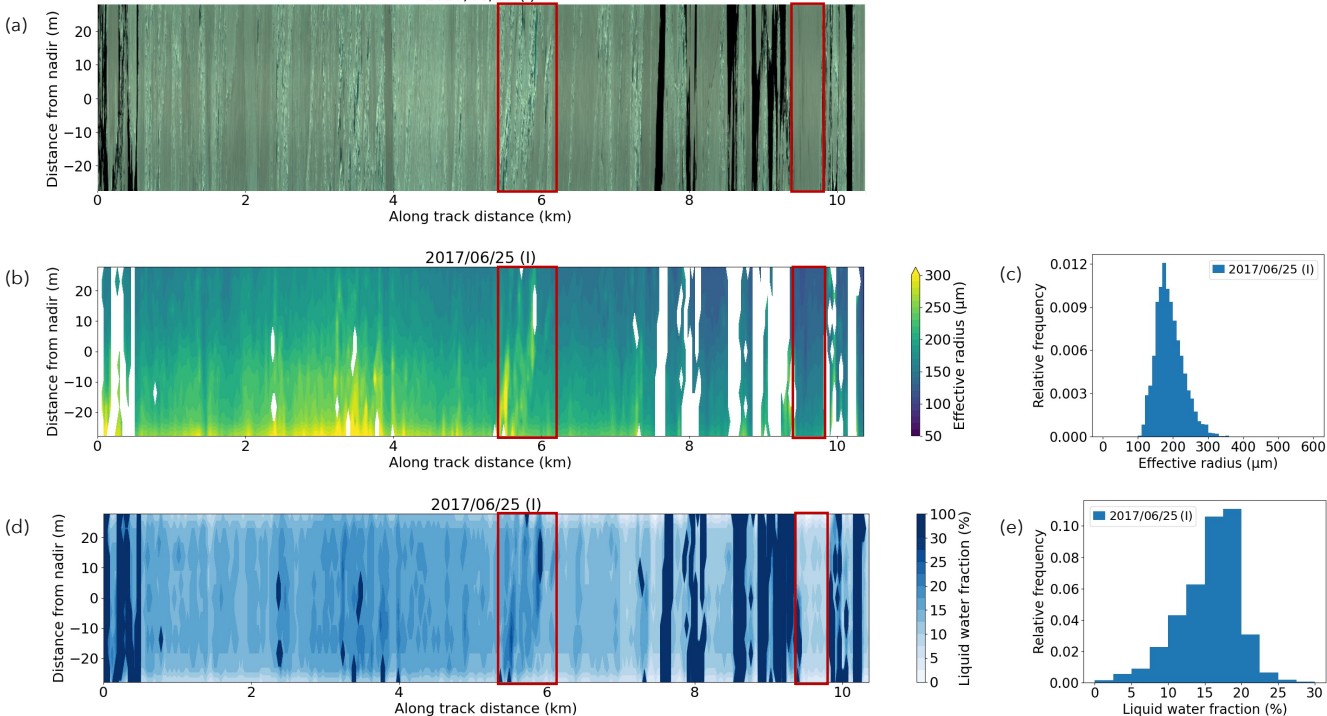

**Figure 5.** Flight section 2017/06/25 (I): **(a)** AisaEagle-RGB-composite. Maps and frequency distributions of **(b)-(c)** $r_{\mathrm{eff}}$, **(d)-(e)** $f_{\mathrm{LW}}$. The maps are plotted over along track distance and distance from nadir. Sections containing melt ponds or open water are excluded and shown in white in **(b)** and dark blue in **(d)**, corresponding to $f_{\mathrm{LW}} = 100\,\%$. For better contrast the colorbar of $f_{\mathrm{LW}}$ is compressed from $30\,\%$ on. Red highlighted areas include specific surface structures as melt ponds and pressure ridges (left), and rather homogeneous snow layer conditions (right).

the derived properties for thirteen observation angles between $-15°$ and $+15°$ converted to distance from nadir on the y-axis and the along track distance on the x-axis.

The $r_{\mathrm{eff}}$-frequency distribution in Fig. 5c shows effective radius values between $100\,\mu\mathrm{m}$ and $350\,\mu\mathrm{m}$ with occurrence of generally higher values towards the south-west (negative distances from nadir) as depicted in the map (Fig. 5b). This $r_{\mathrm{eff}}$-gradient was visible on all south-east or north-west heading flight sections. It might therefore be an effect of geometry, rather

than an actual overall gradient of the effective radius. The nearer the scattering angle towards the forward-scattering peak, the stronger a non-complete representation of the phase function will influence the simulated reflectance spectra. For future application also the influence of different particle shapes on the retrieved $r_{\mathrm{eff}}$ and $f_{\mathrm{LW}}$ should be investigated.

Structures like melt ponds or pressure ridges, visible in the AisaEagle-RGB-composite, are also obvious in the $r_{\mathrm{eff}}$-map as indicated by the left red box. Nevertheless, the reference curves for the $r_{\mathrm{eff}}$-retrieval show a dependence on observation angle

with deviations up to $100\,\mu\mathrm{m}$ between an observation angle of $+15°$ and $-15°$, indicating a sensitivity to surface inhomogeneities and roughness in the vicinity of melt ponds and pressure ridges.





The map and frequency distribution of retrieved snow layer liquid water fraction $f_{\mathrm{LW}}$ presented in Fig. 5d-e show values between $f_{\mathrm{LW}} = 2.5 - 30\,\%$. In order to filter out open water as well as melt ponds from the $r_{\mathrm{eff}}$- and $f_{\mathrm{LW}}$-maps, all spectra that showed unusually high peaks in the wavelength range between $1350\,\mathrm{nm}$ and $2300\,\mathrm{nm}$ due to decreased signal-to-noise ratio were omitted. The filtered out sections are indicated as dark blue areas of $f_{\mathrm{LW}}$ up to $100\,\%$ in the $f_{\mathrm{LW}}$-map.

Overall, a variation of $f_{\mathrm{LW}}$ is visible, with a particularly homogeneous area highlighted in the maps by the right red box. Due to increased uncertainty of the AisaHawk calibration function for larger observation angles, the least square fit showed higher residuals towards the FOV edges, leading to higher uncertainties in these angle ranges.

Table 2 presents an overview of all analyzed cases including the mean, median, and standard deviation for all $r_{\mathrm{eff}}$- and $f_{\mathrm{LW}}$-maps. The retrieved effective radii were mostly between $50 - 600\,\mu\mathrm{m}$ and flight section averages of the order of $100 - 400\,\mu\mathrm{m}$. This is a realistic magnitude compared to findings of particle sizes from Mei et al. (2021) and Jäkel et al. (2021) for an Arctic field campaign in March/April 2018. They derived the snow grain size from measurements of the Sea and Land Surface Temperature Radiometer (SLSTR) instrument onboard Sentinel-3 and airborne SMART albedo measurements, respectively and found snow grain sizes of $100 - 350\,\mu\mathrm{m}$. Since the ACLOUD campaign was conducted in May/June 2017, thus two months later in the melting season, larger grain sizes can be expected due to snow metamorphism.

The retrieved liquid water fractions, averaged over the respective flight sections, were between $8.7\,\%$ and $15.6\,\%$, corresponding to very wet ($8 - 15\,\%$) and soaked ($> 15\,\%$) snow layers according to the international classification for seasonal snow on the ground (Fierz et al., 2009). The overall high liquid water fraction indicates the runoff phase of snow melting for all cases but flight section 2017/06/08 (I). This is in good agreement with the observation of numerous melt ponds during the

**Table 2.** Overview of statistics of the analyzed flight sections (Std. - Standard deviation).

| Flight | | | $r_{\mathrm{eff}}$ (µm) | | | $f_{\mathrm{LW}}$ (%) | | |
|---|---|---|---|---|---|---|---|---|
| Date | Index | Time (UTC) | Mean | Median | Std. | Mean | Median | Std. |
| 2017/05/31 | I | 16:15:45-16:18:47 | 129 | 127 | 26 | 13.5 | 12.5 | 4.1 |
| 2017/05/31 | II | 16:41:41-16:46:14 | 115 | 114 | 18 | 9.7 | 10.0 | 4.0 |
| 2017/06/08 | I | 10:22:46-10:24:10 | 140 | 141 | 16 | 8.7 | 10.0 | 2.9 |
| 2017/06/25 | I | 12:24:32-12:27:24 | 190 | 185 | 39 | 14.4 | 15.0 | 4.1 |
| 2017/06/25 | II | 12:31:52-12:35:24 | 184 | 180 | 34 | 14.8 | 15.0 | 3.2 |
| 2017/06/25 | III | 12:49:02-12:52:39 | 214 | 212 | 39 | 15.6 | 15.0 | 2.3 |
| 2017/06/25 | IV | 14:31:20-14:34:51 | 378 | 368 | 66 | 15.4 | 15.0 | 2.1 |
| 2017/06/25 | V | 15:20:48-15:23:13 | 273 | 267 | 53 | 12.8 | 12.5 | 2.2 |
| 2017/06/25 | VI | 15:25:10-15:28:53 | 283 | 276 | 53 | 13.6 | 15.0 | 2.7 |
| 2017/06/25 | VII | 15:41:41-15:43:37 | 275 | 270 | 52 | 13.2 | 12.5 | 2.2 |
| 2017/06/25 | VIII | 15:58:40-16:02:15 | 288 | 282 | 61 | 12.0 | 12.5 | 2.8 |



flight sections on 25 June 2017 ($f_{\mathrm{LW}} = 12.0 - 15.6\%$) in comparison to almost no melt ponds covered by the flight sections on 31 May 2017 and 08 June 2017 ($f_{\mathrm{LW}} = 8.7 - 13.5\%$). Nevertheless, especially for flight sections with no observable melt ponds, liquid water fractions above $8\%$ could have overestimated the actual snow wetness or be attributed to small or freshly refrozen leads that were not detected as areas of open water.

Since flight sections were selected from three different dates (31 May 2017, 08 June 2017, and 25 June 2017), the temporal
and regional development of the derived parameters was investigated.

Figure 6 shows $r_{\mathrm{eff}}$-maps of three flight sections (a, c, e) and frequency distributions of the data displayed in these maps (b, d, f). An overall increase of $r_{\mathrm{eff}}$ and broadening of the size distribution is visible from flight section 2017/05/31 (II) to 2017/06/25 (IV) and is interpreted to represent the expected snow metamorphism throughout the melting season. However, the derived $r_{\mathrm{eff}}$ seems to depend on the geographical location and local variations. In Fig. 6f the frequency distribution of flight
section 2017/06/25 (IV) is plotted together with the distribution of flight section 2017/06/25 (I) ($r_{\mathrm{eff}}$-map shown in Fig. 5), which was conducted two hours earlier and around $100\,\mathrm{km}$ south-westerly. The two particle size distributions show significant differences, with the 2017/06/25 (I) case consisting of overall smaller particle sizes and a narrower distribution in comparison to the 2017/06/25 (IV) distribution. Both flight sections have similar temporal length and across track coverage. Therefore,

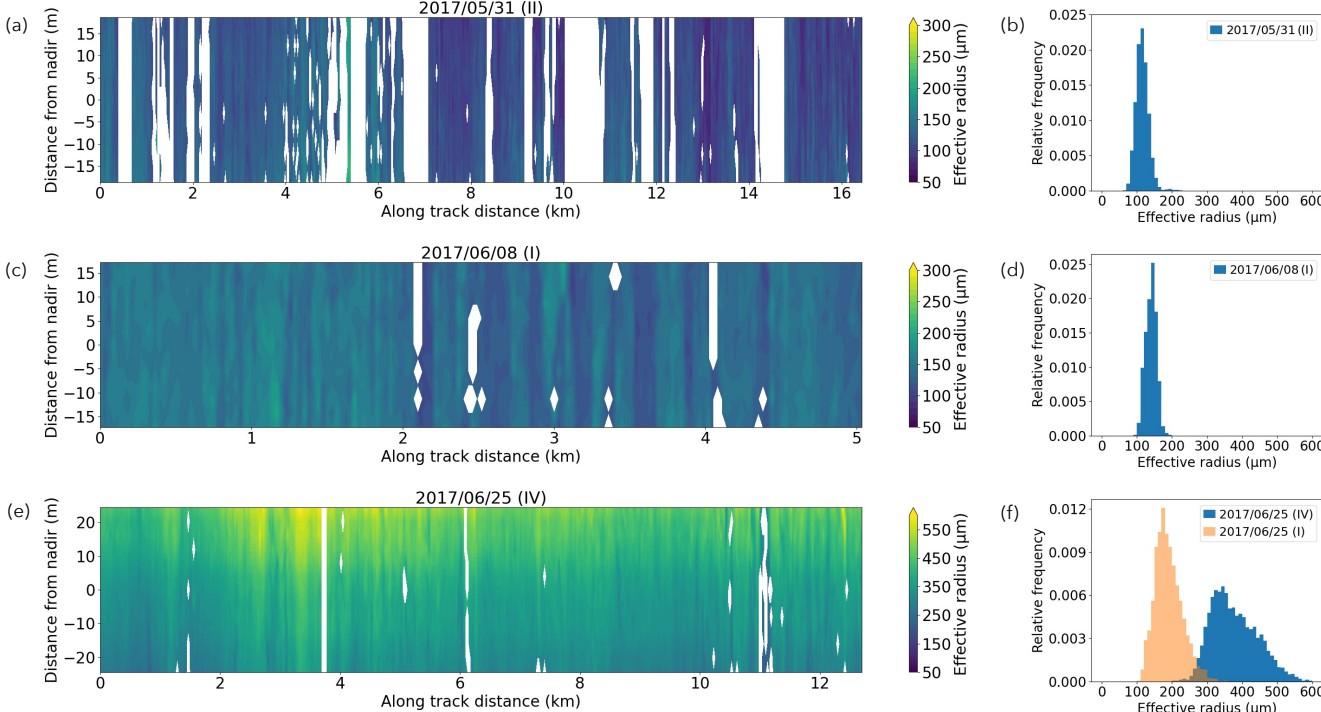

**Figure 6.** Maps of $r_{\mathrm{eff}}$ and respective frequency distributions for flight sections **(a)-(b)** 2017/05/31 (II), **(c)-(d)** 2017/06/08 (I), **(e)-(f)** 2017/06/25 (IV). In case of section 2017/06/25 (IV) the colorbar maxima were adapted to account for overall higher $r_{\mathrm{eff}}$. The $r_{\mathrm{eff}}$-frequency distribution of flight section 2017/06/25 (I) (map shown in Fig. 5) is also shown in **(f)** to represent geographical variability.



differences can be attributed only to local characteristics. Hence, temporal variations are concealed by the seemingly stronger
effects of geographical location.

Figure 7 shows $f_{\mathrm{LW}}$-maps and frequency distributions for the same flight sections that were presented in Fig. 6. Similar to
$r_{\mathrm{eff}}$, also the distribution of $f_{\mathrm{LW}}$ seems to be influenced rather by location than season. Here, the expected increase in mean
$f_{\mathrm{LW}}$ during the season is not represented and even a decrease during the flight section on 08 June 2017 is visible. However,
also in this case the influence of geographical location might again overlay any visible effect of temporal changes throughout
the melting season, since the flight section on 08 June 2017 was carried out further north than the other two (see flight map
in Fig. 2), where lower $f_{\mathrm{LW}}$ could be expected. Figure 7f shows the $f_{\mathrm{LW}}$-distributions of flight sections 2017/06/25 (IV) and
2017/06/25 (I) ($f_{\mathrm{LW}}$-map in Fig. 5). Some geographical variability is apparent, with the $f_{\mathrm{LW}}$-distribution of section 2017/06/25
(IV) being narrower than that of section 2017/06/25 (I). This could also be connected to the higher melt pond fraction of $0.76\,\%$
for flight section 2017/06/25 (I) compared to $0.41\,\%$ for 2017/06/25 (IV), which could indicate a differing melting progress.
However, the effect seems less pronounced compared to the $r_{\mathrm{eff}}$-distribution in Fig. 6f. Also a daily cycle due to undamped
solar radiation in cloud-free conditions could overlay seasonal effects.

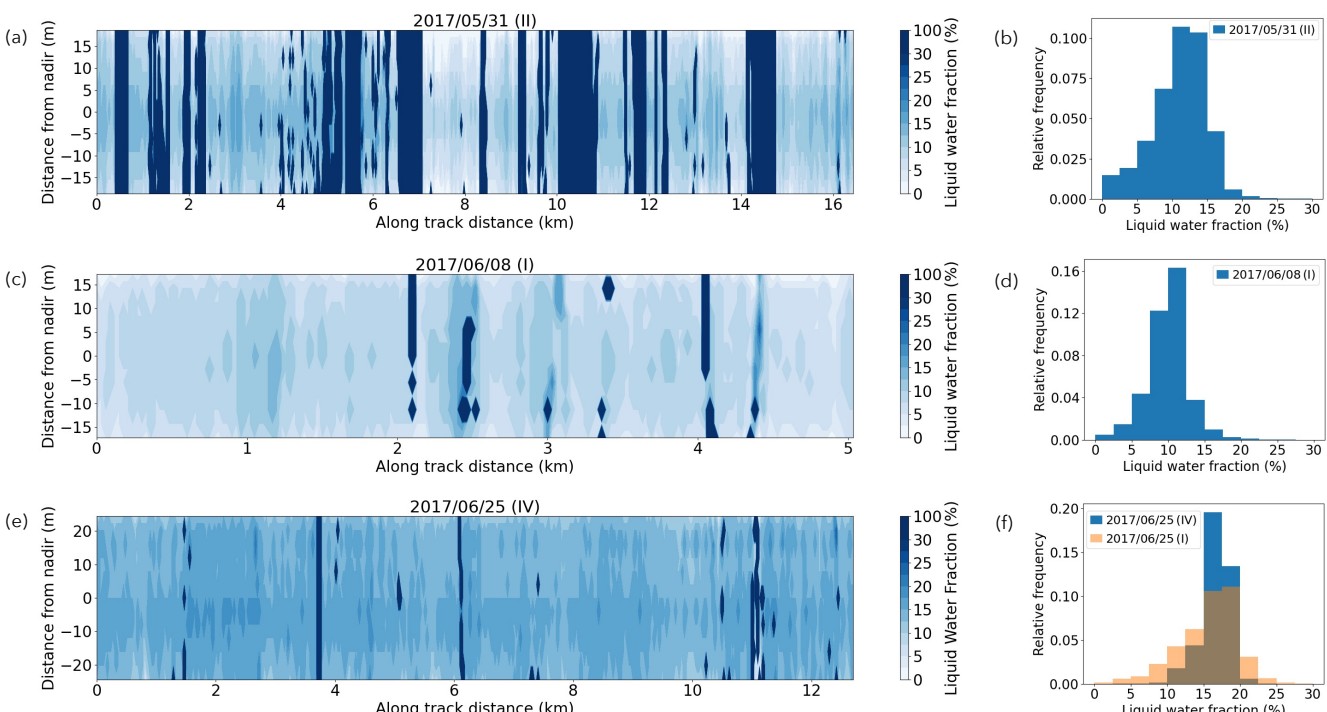

**Figure 7.** Maps of $f_{\mathrm{LW}}$ and respective frequency distributions for flight sections **(a)-(b)** 2017/05/31 (II), **(c)-(d)** 2017/06/08 (I), **(e)-(f)**
2017/06/25 (IV). In case of section 2017/06/25 (IV) the $f_{\mathrm{LW}}$-frequency distribution of flight section 2017/06/25 (I) (map shown in Fig. 5) is
also shown in **(f)** to represent geographical variability.



## 4 Retrieval of melt pond depth

### 4.1 Methodology

The spectral melt pond reflectance is mainly determined by the pond ice bottom reflectance and only limited by the pond depth
(Lu et al., 2016, 2018). To retrieve the melt pond depth, König and Oppelt (2020) analyzed the spectral slopes of log-scaled
simulated reflectance spectra of ponds with different pond ice bottom characteristics and depths at the wavelength $\lambda = 710\,\text{nm}$,
where pond water absorption causes distinct attenuation implying a depth dependence. They found a property that is nearly
independent of the pond ice bottom characteristics and strongly correlated with the pond depth $z$. This relation can be described
by a linear equation:


$$z = a(\theta_{\text{Sun}}) + b(\theta_{\text{Sun}}) \left[ \frac{\partial \log(\mathcal{R}_{\lambda,*})}{\partial \lambda} \right]_{\lambda=710\,\text{nm}}, \tag{3}$$

with $\mathcal{R}_{\lambda,*} = \mathcal{R}_{\lambda} \cdot (\pi\,\text{sr})^{-1}$. The fitting parameters $a$ and $b$ depend on the solar zenith angle $\theta_{\text{Sun}}$ and the melt pond depth $z$
is retrieved in units of cm (König and Oppelt, 2020). Evaluating the accuracy of this linear model, König and Oppelt (2020)
compared retrieved depths to in situ measurements and stated a coefficient of determination of $0.65$. Zhang et al. (2022) also
applied a modified version of the linear model to albedo measurements. In comparison to other approaches they found limited
reliability and pointed out model-based limitations. However, Linhardt et al. (2021) found a reasonable agreement of melt pond
depths retrieved by the linear model with measurements of a ground-based echo sounder within a range up to $1\,\text{m}$ depth. These
measurements were performed during the Multidisciplinary drifting Observatory for the Study of Arctic Climate (MOSAiC)
campaign in 2019/20 (Nicolaus et al., 2022). Furthermore, König et al. (2020) applied the linear model to airborne imaging
spectrometer observations focussing on the comparison of different atmospheric correction techniques as measurements of the
downward irradiance were affected by the operated helicopter. Regarding this study, with the AisaEagle upward radiance and
the SMART downward irradiance both components of the reflectance (see Eq. 1) were measured and used as input for the
linear model to retrieve the depth of selected melt ponds captured during the ACLOUD campaign.

An application of the linear model by König and Oppelt (2020) is constrained by certain assumptions and limitations,
which led to specific criteria for the selection of overflown melt ponds during the ACLOUD campaign. First, the model is
only applicable under cloud-free conditions as clouds would cause deviating in-water pathways due to diffuse incidence.
Also specular reflections at the water surface would be more likely in purely diffuse illumination conditions. That aspect
is of importance, because measurements of the upward radiance performed above the melt pond also capture water surface
reflections. As only the water leaving radiance is of interest to retrieve pond properties, this component has to be minimized
in order to increase the sensitivity to the pond depth. However, observing a stagnant water body within a narrow FOV under
cloud-free conditions can be regarded as optimal conditions for avoiding glint perturbations (Zibordi et al., 2019; Pitarch et al.,
2020). Therefore, specular reflections could be neglected here, as suggested by König and Oppelt (2020). Furthermore, in the
narrow angular range captured by AisaEagle the reflective behavior of ponded ice is almost isotropic (Goyens et al., 2018).



Consequently, all measurements were assumed to represent nadir conditions, although the observation angle varied. Second, pure melt pond water without any dissolved matter is assumed. This way, the depth retrieval is based on water absorption along the traversed pathway through the pond. Due to increasing absorption with depth, König and Oppelt (2020) stated a model applicability for depths reaching a maximum of $1\,\mathrm{m}$. Therefore, only ponds with apparently light blue color were selected to limit the probability of mixing with ocean water. Third, based on the general horizontal plane assumption, flight sections with

aircraft pitch and roll angles exceeding $4.5°$ in absolute values were excluded from the retrieval. Figure 2 shows the selected melt pond locations along the flight track on 25 June 2017. In total five ponds were selected, of which three were overflown consecutively and are depicted by a single orange circle.

To perform the melt pond depth retrieval, the downward irradiance of SMART was interpolated to match the temporal resolution of AisaEagle (see Table 1). This ensured a sufficiently high spatial resolution to resolve single melt pond pixels.

Thus, the pixel size of the AisaEagle measurements determined the minimum resolvable pond size. For a flight altitude of $100\,\mathrm{m}$, aircraft speed of $60\,\mathrm{m\,s^{-1}}$ and $0.05\,\mathrm{s}$ integration time a nadir AisaEagle pixel would cover an area of $0.06\,\mathrm{m} \times 3\,\mathrm{m}$ (across $\times$ along track). Melt ponds were identified with a mask algorithm, which classified the surface into open ocean water, sea ice/snow and melt ponds according to surface typical reflectance spectra. Pond pixel cluster were found with their respective reflectance spectra, which were calculated according to Eq. 1 and spectrally interpolated to $\Delta\lambda = 1\,\mathrm{nm}$.

Furthermore, based on a comparison with *libRadtran* simulations possible atmospheric effects, occurring between surface and flight level, could be neglected. Representing near surface conditions, the determined reflectance spectra were processed as suggested by König and Oppelt (2020). First, a moving mean filter with a window size of $5\,\mathrm{nm}$ was used to smooth the spectra. Second, to obtain the spectral slope at $\lambda = 710\,\mathrm{nm}$ a Savitzky-Golay filter was applied, fitting a second order polynomial to the log-scaled spectrum and determining the first derivative of a $9\,\mathrm{nm}$ window. The slope as well as the solar zenith angle, which

ranged between $57.7°$ and $63.2°$, were inserted into the linear model by König and Oppelt (2020) (Eq. 3) to retrieve the depth of the five selected melt ponds and their depth statistics. The retrieved depth $z$ is defined here as the depth of a single pond pixel, i.e., pixel depth, of which the spectral reflectance was measured.

## 4.2  Retrieval results

In a case study, the depth of the melt pond P1 was retrieved. The pond has a surface area of $225.4\,\mathrm{m^2}$ and is surrounded by

pressure ridges, as shown in Fig. 8a. For each pond pixel the water depth was retrieved with the linear model by König and Oppelt (2020) yielding the pond depth shown in Fig. 8b. The maximum depth of $0.33\,\mathrm{m}$ was derived for the pond center. Pond parts to the right between $30\,\mathrm{m}$ and $45\,\mathrm{m}$ along track distance are mostly shallower with depths varying around $0.2\,\mathrm{m}$. Overall, the melt pond depth is characterized by a high spatial variability and also represents inversely the underlying sea ice relief.

Figure 8c connects depth statistics of the five selected melt ponds P1 to P5 to their contained total meltwater volume. The

already analyzed pond P1 contains the largest meltwater volume of $47.8\,\mathrm{m^3}$ because of its spatial expansion and rather deep parts. The box and whisker plot with median and mean depth points out a rather symmetrically distributed depth. The main fraction of the pixel depths is located within the whisker range. However, a few outliers of shallower pixels occur. Pond P3 stores the second largest volume with $6.0\,\mathrm{m^3}$ despite covering a smaller area of $39.1\,\mathrm{m^2}$ than P5 with an area of $50.6\,\mathrm{m^2}$ and

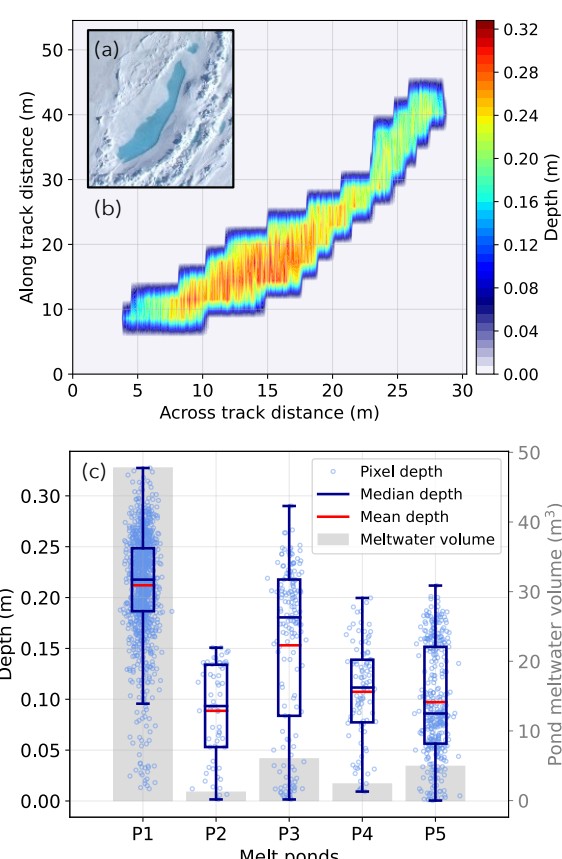

**Figure 8. (a)** Image captured by a digital camera with fisheye lens mounted on *Polar 5*. **(b)** Mapped depth distribution of melt pond P1 according to the pixel sizes in along and across track direction with a colorbar displaying the depth. **(c)** Pixel-based depths (light blue dots) of the five selected melt ponds P1 to P5 plotted together with box (first to third quartile) and whisker ($1.5 \times$ inter-quartile range) plots visualizing the depth distribution with indicated mean (red) and median (dark blue) depth. The grey bars represent the total meltwater volume contained in the respective pond.





a meltwater volume of $4.9\,\mathrm{m}^3$. But a distinct fraction of P3 pond pixels is located at larger depths, resulting also in a skewed

distribution as indicted by median and mean positions. Contrary, the pixel depths of pond P5 rather show a bimodal distribution. Melt pond P4, storing $2.4\,\mathrm{m}^3$, is smaller in surface area with $22.2\,\mathrm{m}^2$ and the depth distribution is nearly symmetrical. In contrast to P1, the smallest pond in terms of area ($13.6\,\mathrm{m}^2$), volume ($1.2\,\mathrm{m}^3$) and, therefore, also depth range is P2. This comparison points out the high variability of the geometrical melt pond dimensions depending on local factors, especially on ice surface roughness and the available amount of meltwater.

As no ground-based reference measurements were performed during the ACLOUD campaign, only in situ pond depth measurements of other campaigns can provide a guideline to evaluate the here retrieved pond depth. The studies by König and Oppelt (2020) and König et al. (2020) obtained reference in situ measurements with a folding ruler on 10 June 2017 with a maximum pond depth about $0.35\,\mathrm{m}$. During the MOSAiC campaign in 2019/20 the sea ice surface observations comprised also melt pond depth measurements in late June and yielded mean depths about $0.1\,\mathrm{m}$ to $0.15\,\mathrm{m}$ (Webster et al., 2022). Therefore,

the magnitude of the here retrieved depths can be assumed to be quite reasonable at the start of the pond evolution.

## 5   Discussion of technical limitations

An estimation of the reliability of the here described retrieval methods is restricted due to the lack of ground-based reference measurements during the ACLOUD campaign. Instead, the derived $r_{\mathrm{eff}}$- and $f_{\mathrm{LW}}$-maps as well as the melt pond depth were compared to typical values from the literature. Additionally, the potential sources of uncertainties and retrieval biases are

quantified and discussed in the following.

### 5.1   Snow layer properties

Considering the different sources of uncertainty imposed by the retrieval approach that add to the uncertainty of the airborne measurements, a deviation from the actual $r_{\mathrm{eff}}$ and $f_{\mathrm{LW}}$ can be expected. In the following, the sources of uncertainty are estimated with sensitivity studies. An overview of uncertainty sources and their contribution is provided in Table 3.

First, the impact of the SMART and AisaHawk measurement uncertainties on the derived parameters was quantified by spectrally adding and subtracting the maximum possible bias (between $5.7\,\% + 3\,\% = 8.7\,\%$ up to $5.7\,\% + 4\,\% = 9.7\,\%$, see Table 1) from the reflectance spectra and then again performing the retrieval approach. This led to deviations of $\Delta r_{\mathrm{eff}} = 4\,\mathrm{\mu m}$ and $\Delta f_{\mathrm{LW}} = 2.5\,\%$, which demonstrate the effectiveness of normalizing the reflectance spectra in order to reduce the influence of systematic errors. Statistical errors were rather small (about $0.1\,\%$) and creating modified reflectance spectra with a Gaussian

error distribution ($\mathrm{Std} = 0.1\,\%$) did not change the derived $r_{\mathrm{eff}}$ and $f_{\mathrm{LW}}$ significantly.

In a similar way, the influence of averaging aircraft height, heading, solar zenith and azimuth angle for the simulations was examined by varying those properties in the simulations between the maximal and minimal value during the flight sections. The retrieval method was performed again for these adapted LUTs leading to the maximal uncertainties in the derived properties that are listed in Table 3. The strongest source of uncertainty are deviations from the aircraft heading due to the sensitivity of

the retrieval method to the phase function of the scattering particles.





**Table 3.** Overview of different sources of uncertainty and their influence on the derived effective radii and liquid water fractions.

| Uncertainty sources | Maximum uncertainty of | |
| --- | --- | --- |
| | $r_{\mathrm{eff}}$ (µm) | $f_{\mathrm{LW}}$ (%) |
| Systematic measurement uncertainty | 4.0 | 2.5 |
| Height variability ($\pm 10.0\,\mathrm{m}$) | 1.0 | 2.5 |
| Heading variability ($\pm 5.0°$) | 50.0 | 5.0 |
| Solar zenith angle variability ($\pm 0.1°$) | 5.0 | 2.5 |
| Solar azimuth angle variability ($\pm 1.0°$) | 7.0 | 2.5 |
| Atmosphere representation | 2.0 | 2.5 |
| Total uncertainty | 69.0 | 17.5 |

Furthermore, the representation of atmospheric conditions in the simulations was analyzed. The simulations, used in the retrieval, were performed assuming a standard Arctic summer atmosphere in combination with radiosonde profiles from Ny-Ålesund for the respective flight day. To evaluate the importance of information on local conditions, additional simulations were performed with the standard atmosphere only. A comparison of both atmosphere representations in the simulations yielded the given deviations for retrieved $r_{\mathrm{eff}}$ and $f_{\mathrm{LW}}$.

The total uncertainty margins, given at the bottom of Table 3, correspond to $50-100\,\%$ of the retrieved $r_{\mathrm{eff}}$ and $f_{\mathrm{LW}}$ showing a high susceptibility of the retrieval method to the examined uncertainty sources. Nevertheless, the total uncertainty of the liquid water fraction might be overestimated, because the retrieved values are restricted to a resolution of $\Delta f_{\mathrm{LW}} = 2.5\,\%$ of the simulations. Increasing the $f_{\mathrm{LW}}$ resolution would probably reduce the total error margin. Moreover, reducing the measurement uncertainty and increasing the wavelength resolution of the measurement devices could further improve the reliability of the retrieval method. In addition to that, only choosing flight sections with very stable headings and only minor changes in solar azimuth and zenith angles (preferably noon or shorter flight sections) would further increase the steadiness of the retrieved parameters.

## 5.2 Melt pond depth

Uncertainties affecting the retrieved pond depth can be ascribed to systematic measurement uncertainties of AisaEagle and SMART in a range of $\pm 3\,\%$ and $\pm 5.7\,\%$, respectively. The total uncertainty of $\pm 8.7\,\%$ was applied to the whole reflectance spectrum. However, the effect on the pond depth was negligible as the linear model by König and Oppelt (2020) is based on the spectral slope of the log-scaled reflectance. An uncertainty of $\pm 2\,\%$ arising from the SMART transfer calibration (Sect. 2.1), which is connected to the temperature dependence of the spectrometer, was applied to vary the steepness of the reflectance spectrum in the spectral range of $9\,\mathrm{nm}$ around $\lambda = 710\,\mathrm{nm}$ that was scanned by the Savitzky-Golay filter. For this uncertainty component a maximum depth deviation of the selected ponds about $\pm 0.07\,\mathrm{m}$ was found and showed a dependence on the





respective solar zenith angle, which was the second input property of the linear model. With increasing solar zenith angle the deviation of the pond depth due to a differing reflectance slope decreased.

Furthermore, also the calculation of the reflectance spectrum slope by the Savitzky-Golay filter itself should be regarded as a

potential uncertainty affecting the retrieval. The filter was applied, as suggested by König and Oppelt (2020), with a polynomial order of 2 and a scanned window size of 9 nm. Thus, at the selected wavelength a polynomial fit was applied to a 9 nm interval of the log-scaled reflectance spectrum and the first derivative, i.e., the slope, was determined. The selection of the window size was based on the compromise between noise removal but preserving important spectral features. In that context a 9 nm window was an adequate choice. However, to quantify the effect of the window size the melt pond depth was also retrieved with a 3 nm

window, i.e., applying no smoothing. The retrieved depth was deviating at maximum about 0.11 m. Therefore, the smoothing is affecting the retrieval distinctly and has to be applied specifically depending on instrument and measurement conditions.

## 6 Conclusion

In this study, snow layer and melt pond properties were retrieved based on airborne imaging spectrometer observations. The retrieval approach for liquid water fraction and effective radius of snow grains is based on a method introduced by Green et al.

(2002), exploiting the spectrally differing absorption indices of ice and liquid water in the near-infrared spectral range. Snow layer reflectance LUTs were simulated for varying liquid water fractions and effective radii to identify the spectral ranges with the strongest sensitivity to both parameters. In the spectral range between 1240 nm and 1295 nm the simulated spectra showed an isolated dependence on the grain size and allowed a derivation of a reference curve for the retrieval of the effective radius. Measured snow reflectance spectra were compared to simulations and the respective liquid water fraction and effective radius

values were determined and mapped for eleven flight sections on three days of the ACLOUD campaign. The flight section averages of retrieved liquid water fractions ranged from 8.7 % to 15.6 % and the effective radii from 115 μm to 378 μm. These results were analyzed in context of temporal snow layer development, but the effect was mainly masked by the geographical location of the measurements. The small number of cloud-free flight sections during the ACLOUD campaign did not allow to average over different flight sections for each day with varying geographical locations and times. Additionally, the total

uncertainty margin of the approach was evaluated by performing sensitivity studies that took uncertainty in measurements and simulations into account. In order to reduce the number of free variables, here only droxtal shaped ice particles were considered in the simulations. Future studies should investigate the effect of different ice particle shapes on the retrieval method. Furthermore, same effective sizes of ice and liquid water particles were assumed in this study. Donahue et al. (2022) used a similar model of same-sized ice and liquid water particles, which compared well to laboratory and field measurements.

However, the actual relation between ice and liquid water particle size is unknown and might also vary with melting regime (Colbeck, 1978, 1979; Hannula and Pulliainen, 2019). It was concluded that a realistic representation of the reflective behavior of a melting snow layer in radiative transfer simulations is crucial for reliable retrieval results.

In the second part of this study, the melt pond depth was retrieved with the linear model developed by König and Oppelt (2020). This approach is almost independent of the pond ice bottom reflectance and is based on the slope of the log-scaled



reflectance spectrum at $710\,\mathrm{nm}$ as well as the solar zenith angle. The pond depth and the in-pond depth distribution were analyzed for five selected cases with a maximum retrieved depth of $0.33\,\mathrm{m}$. It can be stated that the pond depth is a spatially highly variable property. The importance of a precise pond depth retrieval was highlighted estimating the meltwater volumes stored in the observed melt ponds. Uncertainties affecting the retrieval included the measurement uncertainty and retrieval assumptions, comprising pure pond water and negligible water surface reflections. Another aspect concerns the data processing

and especially the smoothing procedure, which can introduce further uncertainties. Also a complete independence of the pond ice bottom reflectance cannot be guaranteed for the linear model, as it was stated by König and Oppelt (2020).

    The two retrieval methods illustrate the potential to study melting processes on sea ice by combining the observed snow grain size, liquid water fraction, and melt pond depth. However, a validation with ground-based reference measurements would be required for further improvements of the approaches and their adjustment to airborne measurements. In future studies, different

areas of sea ice should be overflown multiple times throughout the entire melting season to characterize the temporal development of snow layer composition and melt ponds. This would exploit the full potential of airborne imaging spectrometers, e.g., AisaEagle and AisaHawk, to map the Arctic sea ice surface transition, following the meltwater path from the snow layer to melt ponds.

*Data availability.* The airborne measurements performed during the ACLOUD campaign are published on the PANGAEA database. The

radiances measured by AisaEagle and AisaHawk are available at https://doi.org/10.1594/PANGAEA.902150 (Ruiz-Donoso et al., 2019). The irradiance measurements of the SMART albedometer were published by Jäkel et al. (2019) at https://doi.org/10.1594/PANGAEA.899177.

## Appendix A: Radiative transfer simulations

The spectral downward irradiance and upward radiance were simulated with the library of radiative transfer routines and programs libRadtran (Emde et al., 2016; Mayer et al., 2019). To solve the radiative transfer equation, the discrete ordinate al-

gorithm DISORT (Stamnes et al., 2000) was selected. For the intensity correction the Legendre moments were used (Nakajima and Tanaka, 1988). Furthermore, the extraterrestrial solar spectrum by Gueymard (2004) and the absorption parameterization by Gasteiger et al. (2014) were applied. Atmospheric conditions were described by standard profiles of pressure, temperature, relative humidity, air and trace gas densities for the subarctic summer (Anderson et al., 1986). Additional atmospheric information were provided by radio soundings performed at Ny-Ålesund (Maturilli, 2020).

Further input parameters comprised the flight day and altitude, as well as solar/viewing azimuth and zenith angles describing the sun position/observation geometry with respect to the aircraft heading in order to simulate reflectances comparable to the pushbroom imaging spectrometer measurements.

    To represent the snow layer, a mixed-phase cloud layer located at $0-1\,\mathrm{m}$ above the surface was defined by a constant total water content $\mathrm{TWC} = 100{,}000\,\mathrm{g\,m^{-3}}$ while varying the liquid water and ice water content to account for melting processes.

An external mixture of liquid water and ice particles was assumed (Donahue et al., 2022). The extinction coefficient, the



single scattering albedo, and the scattering phase function for a gamma size distribution of liquid water spheres and smooth ice droxtals were calculated with the Mie-tool (Wiscombe, 1980), provided by libRadtran, and the Yang tables (Yang et al., 2000), respectively. These properties were derived with 2048 Legendre moments and $\delta$-M-scaling (Wiscombe, 1977) to ensure an adequate resolution of the phase function.

*Author contributions.*   SR and CL performed the radiative transfer simulations, worked on the retrievals, and drafted the article. EJ initiated this study and processed the SMART data. MS processed the AisaEagle and AisaHawk data. AE and MW designed the experimental basis of this study. All authors contributed to the editing of the article and to the discussion of the results.

*Competing interests.*   Manfred Wendisch is a member of the editorial board of Atmospheric Measurement Techniques.

*Acknowledgements.*   We gratefully acknowledge the funding by the Deutsche Forschungsgemeinschaft (DFG, German Research Foundation)
– Project Number 268020496 – TRR 172, within the Transregional Collaborative Research Center "ArctiC Amplification: Climate Relevant Atmospheric and SurfaCe Processes, and Feedback Mechanisms (AC)³".



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
