# Peer review of "Retrieval of snow layer and melt pond properties on Arctic sea ice from airborne imaging spectrometer observations"

_Atmospheric Measurement Techniques, 2023_

## Author Comment (AC1)

Answers to comments of Referee #1

We want to thank the referee for the comments to our submitted manuscript.

Our replies are structured as follows:

*(1) Referee comment*

(2) Author response

**(3) Manuscript changes** (given line numbers refer to the revised manuscript)

(4) References are listed in case they were not included in the original manuscript

*I have only one critical comment on the manuscript:*

*The statements "the stronger a non-complete representation of the phase function will influence the simulated reflectance spectra" and "[the] sensitivity of the extraction method to the phase function" is one of the "strongest sources of uncertainty" are incorrect. In the case of multiple scattering, the details of the single scattering phase function are not important and in problems such as the one considered it is sufficient to use the so-called transport approximation with the correct value of the asymmetry factor of scattering*

*I would like to recommend the following literature on this subject:*

*- L.A. Dombrovsky, The use of transport approximation and diffusion-based models in radiative transfer calculations, Computational Thermal Sci. 4 (4) (2012) 297–315. http://doi.org/10.1615/ComputThermalScien.2012005050*

*- L.A. Dombrovsky and A.A. Kokhanovsky, Solar heating of the cryosphere: Snow and ice sheets, Ch. 2 in the book "Springer Series in Light Scattering", edited by A. Kokhanovsky, Springer Nature, 2021, v. 6, pp 53-109. https://doi.org/10.1007/978-3-030-71254-9_2*

*Please consider including these articles in the reference list.*

We assume that we did not clearly indicate the characteristics of the measured quantities in the manuscript. The measurements used in the retrieval are the spectral upward radiance (unit: W m$^{-2}$ nm$^{-1}$ sr$^{-1}$) and the spectral downward irradiance (unit: W m$^{-2}$ nm$^{-1}$). In our opinion, the angular information of the scattering phase function can only be negligible, if hemispherical radiative properties are regarded as it would be the case for the irradiance. Contrary, the radiance has a directional dependence. Furthermore, low sun conditions are frequent in the Arctic and imply decreasing penetration depths into the snow layer. This results in an increased reflection and more single scattering (e.g., Warren, 1982). This introduces an angular distribution of the reflected radiance impacted by the scattering phase

function. Additionally, the scattering phase function determines the dependence of the reflected radiance on the solar zenith angle (e.g., Carlsen et al., 2017). The same dependencies show up in our radiative transfer simulations suggesting that the angular information of the scattering phase function cannot be neglected. The left figure below shows the simulated normalized reflectivity for fixed effective radius and liquid water fraction and varying viewing zenith angle (VZA) with 0° as nadir and ±15° indicating the sensor pixels left and right from the central nadir pixel (a schematic illustration of the observation geometry is given below). The right figure considers an additional change of the aircraft heading with ±5°, which is affecting the viewing azimuth angle (VAA) that is determined by the aircraft heading ±90°. For both analyses an angular dependence of the reflectivity is apparent, especially regarding a variation of the aircraft heading and, therefore, viewing azimuth angle. Consequently, for the upward radiance used in our study we need to refer to the scattering phase function and its implications for the retrieval.

[Figure]

[Figure]

**Observation Geometry – AisaEagle and AisaHawk**

o viewing zenith angle VZA

o viewing azimuth angle VAA

[Figure]

[Figure]

References:

Carlsen, T., Birnbaum, G., Ehrlich, A., Freitag, J., Heygster, G., Istomina, L., Kipfstuhl, S., Orsi, A., Schäfer, M., and Wendisch, M.: Comparison of different methods to retrieve optical-equivalent snow grain size in central Antarctica, The Cryosphere, 11, 2727–2741,https://doi.org/10.5194/tc-11-2727-2017, 2017.

Warren, S. G.: Optical properties of snow, Rev. Geophys., 20, 67–89, https://doi.org/10.1029/RG020i001p00067, 1982.

---

## Author Comment (AC2)

We want to thank the referee for providing constructive feedback and comments, which significantly helped to improve our submitted manuscript.

Our replies are structured as follows:

*(1) Referee comment*

(2) Author response

**(3) Manuscript changes** (given line numbers refer to the revised manuscript)

(4) References are listed in case they were not included in the original manuscript

*The article demonstrates airborne imaging spectroscopy retrievals over arctic sea ice. The retrievals include snow grain size, liquid water fraction, and ice pond depth. Generally, the methodology and retrievals presented are of interest to the cryosphere remote sensing community. However, there are several major items that need to be addressed before publication, listed here in no particular order:*

1. *Throughout the paper ambiguous remote sensing terminology is used, e.g., "reflectance". Refer to Schaepman-Strub et al (2006) and use correct terminology throughout. For example, the airborne spectrometer measures HDRF/HCRF while the paired albedometers measure BHR. Additionally, more details about the radiative transfer simulations need to be provided. Are the simulations representative of white or black sky albedo or HDRF? How is the illumination and observation angle geometry being handled in the radiative transfer modeling?*

We realized that the used terminology was unprecise and caused confusion. Following Schaepman-Strub et al. (2006), the ratio of measured upward radiance and downward irradiance can be defined as an angular part of the HDRF (Hemispherical-Directional Reflectance Factor). However, our measurements of the upward radiance do not cover the whole lower hemisphere as the imaging spectrometers have a very narrow field of view of 36°. That is why we want to refer to the terminology that is well-established in the spaceborne remote sensing community and replaced "reflectance" by "reflectivity". The term and quantity "reflectivity" considers the normalization of the upward radiance by the downward irradiance and expresses that only a limited angular range close to nadir is observed. In case of other reflection properties and definitions applied in other studies, we refer to these as general "reflection" quantities to avoid confusion when citing those studies. We only write "reflectivity" in context of our measurements.

Lines 64-68:

"An airborne imaging spectrometer measured the spectral upward radiance $I_\lambda^\uparrow$ (W m$^{-2}$ nm$^{-1}$ sr$^{-1}$) in a narrow angular range close to nadir, which is normalized by the spectral downward irradiance $F_\lambda^\downarrow$

(W m$^{-2}$ nm$^{-1}$), measured by an albedometer, to determine the spectral reflectivity $\mathcal{R}_\lambda$ of the Arctic sea ice surface according to:

$$\mathcal{R}_\lambda = \frac{\pi \cdot I_\lambda^\uparrow}{F_\lambda^\downarrow} \ \mathbf{sr} \ ."$$

Regarding the simulations: We simulated the property "reflectivity" as defined in our response above. Atmospheric scattering and absorption are considered in the simulations.

Observation geometry: The Figure below illustrates the observation geometry of the imaging spectrometers AisaEagle and AisaHawk. The viewing zenith angle is determined by the regarded pixel and its angular distance from nadir. The viewing azimuth angle is depending on the aircraft heading and the regarded sensor pixel located ±90° from the heading.

[Figure]

In the revised manuscript we now provide more details on that in Sect. 3.1 as well as in the Appendix A and Table A1. In addition, we added a schematic illustration of flight direction and Sun position for each $r_{\mathrm{eff}}$- and $f_{\mathrm{LW}}$-map shown in Sect. 3.2.

Lines 155-157:

"The observation geometry in the simulations was indicated via viewing azimuth angle (aircraft heading ±90°) and viewing zenith angle of the imaging spectrometers in order to consider the correct viewing geometry relative to the Sun position. Further information are provided in Table A1."

Appendix A and Table A1:

Lines: 415-432:

"Appendix A: Radiative transfer simulations

The spectral downward irradiance and upward radiance were simulated with the library of radiative transfer routines and programs *libRadtran* (Emde et al., 2016; Mayer et al., 2019). To solve the radiative transfer equation, the discrete ordinate algorithm DISORT (Stamnes et al., 2000) was selected. For the intensity correction the Legendre moments were used (Nakajima and Tanaka, 1988). Furthermore, the

extraterrestrial solar spectrum by Gueymard (2004) and the absorption parameterization by Gasteiger et al. (2014) were applied. Atmospheric conditions were described by standard profiles of pressure, temperature, relative humidity, air and trace gas densities for the subarctic summer (Anderson et al., 1986). Additional atmospheric information were provided by radio soundings performed at Ny-Ålesund (Maturilli, 2020).

Further input parameters are listed in Table A1 and comprised the flight day and altitude, as well as solar/viewing azimuth and zenith angles describing the Sun position/observation geometry with respect to the aircraft heading in order to simulate reflectivities comparable to the pushbroom imaging spectrometer measurements.

To represent the snow layer, a mixed-phase cloud layer located at $0 - 1$ m above the surface was defined by a constant total water content TWC = 100,000 g m$^{-3}$ while varying the liquid water and ice water content to account for melting processes. An external mixture of liquid water and ice particles was assumed (Donahue et al., 2022). The extinction coefficient, the single scattering albedo, and the scattering phase function for a gamma size distribution of liquid water spheres and smooth ice droxtals were calculated with the Mie-tool (Wiscombe, 1980), provided by *libRadtran*, and the Yang tables (Yang et al.,2000), respectively. These properties were derived with 2048 Legendre moments and δ-M-scaling (Wiscombe, 1977) to ensure an adequate resolution of the phase function."

**Table A1.** Aircraft orientation and illumination conditions during the selected flight sections given with their respective standard deviation.

| Flight | | Aircraft | | Solar | |
| --- | --- | --- | --- | --- | --- |
| Date | Index | Altitude (m) | Heading (°) | Azimuth angle (°) | Zenith angle (°) |
| 2017/05/31 | I | $90.7 \pm 19.0$ | $172.6 \pm 1.6$ | $257.2 \pm 0.2$ | $65.63 \pm 0.03$ |
| 2017/05/31 | II | $69.6 \pm 5.6$ | $53.5 \pm 2.3$ | $264.3 \pm 0.5$ | $66.86 \pm 0.08$ |
| 2017/06/08 | I | $64.2 \pm 5.0$ | $338.5 \pm 0.5$ | $165.6 \pm 0.1$ | $59.52 \pm 0.01$ |
| 2017/06/25 | I | $103.8 \pm 4.7$ | $307.4 \pm 1.2$ | $199.1 \pm 0.1$ | $57.67 \pm 0.02$ |
| 2017/06/25 | II | $131.1 \pm 4.8$ | $138.7 \pm 1.0$ | $201.2 \pm 0.4$ | $57.78 \pm 0.01$ |
| 2017/06/25 | III | $81.3 \pm 2.4$ | $53.1 \pm 1.3$ | $207.4 \pm 0.5$ | $58.25 \pm 0.05$ |
| 2017/06/25 | IV | $91.2 \pm 2.6$ | $145.3 \pm 1.8$ | $239.3 \pm 0.4$ | $61.93 \pm 0.04$ |
| 2017/06/25 | V | $63.9 \pm 3.2$ | $147.0 \pm 1.7$ | $253.8 \pm 0.3$ | $63.89 \pm 0.03$ |
| 2017/06/25 | VI | $78.5 \pm 3.4$ | $314.0 \pm 0.7$ | $254.9 \pm 0.1$ | $64.06 \pm 0.02$ |
| 2017/06/25 | VII | $87.4 \pm 1.9$ | $147.9 \pm 3.1$ | $259.0 \pm 0.2$ | $64.63 \pm 0.03$ |
| 2017/06/25 | VIII | $154.3 \pm 3.3$ | $148.2 \pm 1.6$ | $263.2 \pm 0.4$ | $65.25 \pm 0.06$ |

Schematic illustration of flight direction and Sun position (see Fig. 4-6), exemplarily for flight section 2017/06/25 (I) according to aircraft heading and solar azimuth angle given in Table A1:

[Figure]

References:

Schaepman-Strub, G., Schaepman, M., Painter, T., Dangel, S., and Martonchik, J.: Reflectance quantities in optical remote sensing - definitions and case studies, Remote Sens. Environ., 103, 27–42, https://doi.org/10.1016/j.rse.2006.03.002, 2006.
* * *
2. *There are 2 grain size retrieval methods presented (figure 3 and 4), however only results from one of the methods (figure 4) is presented. I am confused by this, where are grain size results from the methodology presented in Figure 3? Otherwise, the least-square fitting methodology for grain size should be removed from the manuscript.*
3. *The grain size/LWC methodology presented is based on Green et al 2002, however the LWC and grain size retrievals are decoupled instead of retrieved simultaneously. Some discussion around the reasoning for this and its implications should be included. For example, the least squares methodology for grain size (using Parts 1-3) seems to include radiative transfer simulations that are based on wet and dry snow, though only the grain size is being retrieved. What uncertainties does this introduce? Would it be possible find a least square spectra that was simulated using a 200 μm grain size and 10% water and alternatively be very similar to 250 μm and 0% water?*

The least square fit retrieval using Parts 1-3 should in theory retrieve effective radius and liquid water fraction simultaneously. However, due to a limited resolution of the simulations, the results were not sufficiently reliable and alternative retrieval methods (Least Square Fit to Part 1 for $f_{LW}$ and reference curve for $r_{eff}$) were included in the original manuscript. In the revised manuscript, we improved the resolution of the library of simulated reflectivity spectra by interpolating between the simulated reflectivity spectra. This way, the effective radius was retrieved in a resolution of 1 μm. Moreover, this interpolation of simulated reflectivity spectra allowed to also retrieve the liquid water fraction from a least square fit to Parts 1-3 rather than only using Part 1. This way, liquid water fraction and snow effective radius were retrieved from the same least square fit to Parts 1-3, which made the other retrieval methods introduced beforehand obsolete.

In the revised manuscript we removed the retrieval using the reference curve to retrieve $r_{eff}$ and the least square fit using only Part 1 for the retrieval of $f_{LW}$ and adjusted the description of the combined $r_{eff}$- and $f_{LW-}$ retrieval.

A comparison of both methods is shown here with the difference original approach (decoupled) minus revised approach (coupled):

[Figure]

Lines 162-163:

"Moreover, for each $f_{LW}$-step the 16 simulated spectra of varying $r_{eff}$ were transferred to a resolution of $\Delta r_{eff} = 1$ µm by cubic interpolation."

Lines 174-177:

"For the coupled retrieval of $r_{eff}$ and $f_{LW}$, three wavelength ranges were selected for the least square fit (Fig. 3): $\lambda = 982 - 1054$ nm (Part 1), $\lambda = 1181 - 1240$ nm (Part 2), and $\lambda = 1294 - 1320$ nm (Part 3), omitting areas with strong atmospheric absorption. Part 1 covers the reflectivity minimum for pure ice at 1030 nm making it sensitive to $f_{LW}$, while Parts 2 and 3 cover a spectral region that shows a strong dependence on $r_{eff}$ with only minor sensitivity to $f_{LW}$."

Updated statistics in Table 2 for the revised approach:

**Table 2.** Overview of statistics of the analyzed flight sections (Std. - Standard deviation).

| Flight | | | $r_{eff}$ (µm) | | | $f_{LW}$ (%) | | |
|---|---|---|---|---|---|---|---|---|
| Date | Index | Time (UTC) | Mean | Median | Std. | Mean | Median | Std. |
| 2017/05/31 | I | 16:15:45-16:18:47 | 147 | 144 | 30 | 10.4 | 10.0 | 3.9 |
| 2017/05/31 | II | 16:41:41-16:46:14 | 129 | 128 | 20 | 9.1 | 10.0 | 2.6 |
| 2017/06/08 | I | 10:22:46-10:24:10 | 158 | 159 | 18 | 6.5 | 7.5 | 2.1 |
| 2017/06/25 | I | 12:24:32-12:27:24 | 207 | 201 | 42 | 16.2 | 17.5 | 3.4 |
| 2017/06/25 | II | 12:31:52-12:35:24 | 201 | 197 | 38 | 16.1 | 15.0 | 2.6 |
| 2017/06/25 | III | 12:49:02-12:52:39 | 235 | 232 | 43 | 16.7 | 17.5 | 2.0 |
| 2017/06/25 | IV | 14:31:20-14:34:51 | 414 | 404 | 72 | 17.3 | 17.5 | 1.7 |
| 2017/06/25 | V | 15:20:48-15:23:13 | 305 | 298 | 59 | 14.3 | 15.0 | 1.7 |
| 2017/06/25 | VI | 15:25:10-15:28:53 | 315 | 308 | 59 | 15.1 | 15.0 | 2.2 |
| 2017/06/25 | VII | 15:41:41-15:43:37 | 308 | 302 | 59 | 14.8 | 15.0 | 1.7 |
| 2017/06/25 | VIII | 15:58:40-16:02:15 | 318 | 311 | 66 | 14.7 | 15.0 | 2.2 |

Updated uncertainty analysis in Table 3 with added root mean square error (RMSE) to give more information on the overall deviation:

**Table 3.** Overview of different sources of uncertainty and their influence on the derived effective radii and liquid water fractions given as maximum uncertainty and RMSE (root mean square error).

| Uncertainty sources | Maximum uncertainty of | | RMSE of | |
|---|---|---|---|---|
| | $r_{\mathrm{eff}}$ ($\mu$m) | $f_{\mathrm{LW}}$ (%) | $r_{\mathrm{eff}}$ ($\mu$m) | $f_{\mathrm{LW}}$ (%) |
| Systematic measurement uncertainty | 8.0 | 2.5 | 3.7 | 2.1 |
| Height variability ($\pm 10.0\,$m) | 8.0 | 2.5 | 0.9 | 0.3 |
| Heading variability ($\pm 5.0°$) | 55.0 | 5.0 | 5.8 | 1.0 |
| Solar zenith angle variability ($\pm 0.1°$) | 14.0 | 2.5 | 1.5 | 0.5 |
| Solar azimuth angle variability ($\pm 1.0°$) | 7.0 | 2.5 | 0.6 | 0.3 |
| Atmosphere representation | 8.0 | 2.5 | 1.0 | 0.4 |
| Total uncertainty | 100.0 | 17.5 | 13.5 | 4.6 |

4. *Line 64 – 68: This is somewhat of an overstatement. All three properties are not combined in a single flight line, though if this is possible and demonstrated it would greatly enhance the paper results. Furthermore, the retrievals individually (grain size and water content) have been demonstrated using airborne imaging spectroscopy (e.g., Bohn et al. 2021).*

We agree with the reviewer that the applied retrieval approaches are not combined for the snow layer properties and the melt pond depth and were only partly applied to the same flight sections. However, we aim to emphasize that a simultaneous retrieval of all three properties (snow layer liquid water fraction, snow grain size, melt pond depth) for the same observed scene is possible and is at least exemplarily shown in our study for the flight section 2017/06/25 (I), which is analyzed in terms of $f_{\mathrm{LW}}$ and $r_{\mathrm{eff}}$ in the maps of Fig. 4 and is also covering (left red box) the melt pond P1 shown in Fig. 7a. We changed and added the following lines in the manuscript:

Lines 69-73:

"The Arctic sea ice conditions in spring allowed to observe the snow layer and melt ponds simultaneously. Our work comprises the adaptation and application of the approaches by Green et al. (2002) (retrieval of snow layer liquid water fraction and snow grain size) and König and Oppelt (2020) (retrieval of melt pond depth) for selected case studies. These analyses can provide a basis for future airborne observations with the aim to determine a combined picture of the snow layer and melt pond evolution during the melting season."

Lines 301-302:

"In a case study, the depth of the melt pond P1 was retrieved, which was also covered during flight section 2017/06/25 (I) shown in Fig. 4 inside the left red box."

We further added Bohn et al. (2021) to our literature review in the introduction:

Lines 36-38:

"Bohn et al. (2021) developed a methodology to retrieve snow grain size, liquid water fraction, and the mass mixing ratio of light absorbing particles from spectral reflection measurements with optimal estimation for airborne and spaceborne applications."

References:

Bohn, N., Painter, T. H., Thompson, D. R., Carmon, N., Susiluoto, J., Turmon, M. J., Helmlinger, M. C., Green, R. O., Cook, J. M., and Guanter, L.: Optimal estimation of snow and ice surface parameters from imaging spectroscopy measurements, Remote Sens. Environ., 264, 112 613, https://doi.org/https://doi.org/10.1016/j.rse.2021.112613, 2021.
* * *
5. *Consider reorganizing the paper such that the methods are all presented together followed by the results. The current organization is hard to follow as a reader because it jumps around from methods to results. Additionally, the paper would greatly benefit from further editing to make the paper more concise and flow better.*

We appreciate your feedback. In the process of creating the manuscript, we changed the paper structure several times, each version having its (dis-)advantages. The current version seems to be the best compromise of all versions since the methodology differs for the retrieval of snow layer properties on the one hand and melt pond depth on the other. By combining approach methodology and respective results, the potential and limitations become more comprehensible. Therefore, we would like to keep the current organization.
* * *
*Line 39-40 is the grain "size" a radius of diameter, should be defined?*

We changed that line accordingly.

Lines 38-40:

"Jäkel et al. (2021) compared optical equivalent snow grain radius retrieval methods based on the grain size-dependent absorption in the solar spectral range, which were applied to ground-based, airborne, and spaceborne reflection measurements."
* * *
*Line 49: Green tested outside using a block of snow under natural solar conditions, not in a laboratory.*

Thank you for the comment, we clarified that now in the manuscript.

Lines 51-52:

"This approach was tested on a snow sample block in the field under cloud-free solar illumination by Green et al. (2002) and validated by Donahue et al. (2022) with further field and laboratory experiments."
* * *
*Line 54-56: Unclear, sentence needs to be restated.*

We restated that sentence.

Lines 56-59:

"Typically, the depth of melt ponds on sea ice reaches at maximum 1 m and is depending on the local meltwater availability and surface topography. Multi-year ice is usually characterized by surface ridges and depressions providing vertically more extended basins for deeper melt ponds compared to often level first-year ice surfaces, on which shallower ponds form (Untersteiner, 1961; Morassutti and LeDrew, 1996; Konig et al., 2020; Webster et al., 2022)."
* * *
*Line 63: "In this study" is repetitive since the paragraph started with these 2 sentences prior.*

Thank you for the comment. We changed "In this study" to "Our work" and restated the sentence, which we now already mentioned in our response to your comment 4. Please refer to our response above.
* * *
*Table 1: Under SMART should there be an F (upward arrow) as well? Line 81-82 states that it measures upward and downward looking irradiance.*

This is correct, SMART measures both. However, Table 1 should give an overview of measured properties that were applied in the retrievals. Thus, only the downward irradiance measured by SMART is mentioned. We clarified that in the caption of Table 1:

"Description of measured quantities that were applied in the retrievals by characterizing the respective instrument [...]"

*Line 103: Example of ambiguous usage of "reflectance". More details on the simulations should be included. How was the observation and illumination angle geometry represented in the modeling?*

We want to refer to our response to your first comment.

*Line 139: How was this smoothing done?*

Thanks for the comment, we changed the sentence to clarify.

Lines 145-146:

"For the retrieval of $r_{eff}$ and $f_{LW}$, the AisaHawk measurements (20 Hz resolution) were averaged to fit the SMART measurements (2 Hz resolution)."

*Line 152: observation and illumination conditions?*

We describe the considered conditions now in more detail in Sect. 3.1 and Table A1 in the Appendix, and want to refer to our response to your first comment.

*Line 158-161: Does this normalization change the absolute magnitude of the reflectance spectrum? How might this impact the grain size retrieval? This should be mentioned in the discussion.*

We clarified and justified the normalization in the revised manuscript:

Lines 167-172:

"In order to reduce the influence of the wavelength-dependent systematic errors in the instrumental calibration, measured and simulated reflectivity spectra were normalized by the measured or respectively simulated reflectivity value at the wavelength $\lambda = 1100$ nm, where the absorption indices of liquid water and ice are almost identical. Hence, the information on effective radius and liquid water fraction is represented by the spectral shape of the normalized reflectivities rather than by their absolute values. Therefore, the normalization enables a more distinct separation of the sensitivity to both properties in the regarded wavelength ranges."

*Figure 3. This figure could be improved. Consider showing a few selected simulations that include the full spectra and one or two measurements that match. You can highlight the regions (Part 1,2,3) that are used for the retrieval with vertical shading bars.*

We updated Fig. 3 according to your suggestions. However, regarding the measurements we do not show the full spectral range. Water vapor absorption bands are excluded as the incoming solar radiation is weak in these spectral ranges and the derived reflectivity highly uncertain (low signal-to-noise-ratio).

[Figure]

*Figure 5. (a) Why is the RBG true color image green? I would expect it to be white.*

Figure 5a was not intended to be an RGB true color image as it was not spectrally weighted accordingly. We clarified that in the caption of this Figure in the revised manuscript (now Fig. 4):

**" (without spectral weighting for true color impression) "**

*Figure 5. A scale bar and north arrow should be added to retrieval maps.*

We now added an illustration showing the flight direction and Sun position for each selected flight section in Fig. 4 to 6. We want to refer to our response to your first comment.

*Line 185-187: Consider moving to discussion section.*

We are aware of the interference of result description and discussion in these lines. However, we wanted to combine both to explain error sources next to the maps, which show these angular dependencies. Therefore, we would like to leave the position of these lines unchanged.
* * *
*Line 200-202 and Table 2: Generally, the grain sizes retrieved seem low for water contents that are mostly >10%. For example, Green 2002 retrieved 10% LWC and grain radius of 550 um. Consider adding some discussion around this topic. How might the grain shape chosen influence the grain size retrieval. Further, How does decoupling the grain size and LWC retrieval effect the grain size retrieval?*

Thank you for the comment, as we cannot provide any reference measurements, the comparison to other studies is essential and as a guideline helpful. However, requirements for an adequate comparison comprise similar environmental conditions, in which a snow layer is observed. The airborne measurements, regarded in our study, were carried out in the Arctic and the snow layer on the sea ice was affected by the specific regional conditions, e.g., sea ice surface characteristics, snowfall, snow ageing processes. In contrast, Green et al. (2002) analyzed a snow sample, which was put into a freezer and afterwards melted under field conditions. This is a useful procedure to evaluate the retrieval approach and the sensitivity to the amount of liquid water contained in snow, but the comparability to our measurements is limited. Therefore, a direct comparison to the results of Green et al. (2002) would not be based on the same boundary conditions. Nevertheless, we compared the retrieved effective radii to results by Mei et al. (2021) and Jäkel et al. (2021) of another Arctic field campaign (original manuscript: lines 201-205, revised manuscript: lines 204-208).

However, we also took further aspects into consideration to evaluate the retrieval results. From the $f_{LW}$ perspective, these high liquid water fractions could be justified by the following statement:

Lines 217-219:

**"However, since the approach is mostly sensitive to the uppermost snow layers, high retrieved $f_{LW}$-values could be explained by the daily melting cycle rather than due to an overall soaked snowpack."**

Regarding the grain shape: Many studies state a clear relation between reflection of a snow layer and the snow grain shape, which affects the scattering phase function (e.g., Picard et al., 2009; Dang et al., 2016). However, impacts of the selection of grain shape have to be quantified specifically for the observed reflection quantity and the retrieval approach. Consequently in a next step, liquid water fraction and effective radius would have to be retrieved based on LUTs simulated for different grain shapes to provide an adequate estimate for the shape effect. We decided to simulate reflectivity spectra for grains with droxtal shape justified by the assumption of snow metamorphism processes (original manuscript: lines 118-119, revised manuscript: lines 123-124). In the end, this assumption should be

validated by in-field reference measurements and a sensitivity study concerning the grain effect would be another crucial aspect for future studies.

The (de)coupling issue is addressed in our response to your comments 2 and 3.

References:

Dang, C., Fu, Q., and Warren, S. G.: Effect of Snow Grain Shape on Snow Albedo, J. Atmos. Sci., 73, 3573–3583, https://doi.org/10.1175/jas-d-15-0276.1, 2016.

Picard, G., Arnaud, L., Domine, F., and Fily, M.: Determining snow specific surface area from near-infrared reflectance measurements: Numerical study of the influence of grain shape, Cold Reg. Sci. Technol., 56, 10–17, https://doi.org/10.1016/j.coldregions.2008.10.001, 2009.
* * *
*Figure 7. It is hard to discern the difference between 30% and 100%. Consider removing 100% water fraction from the scale bar or making 100% a separate color and call it "open water", or something similar.*

We updated the liquid water fraction maps in Fig. 4 to 6 and decided to use black instead of blue for indicating the range 30 % - 100 %. Exemplarily for Fig. 6a:

---

## Author Comment (AC3)

Answers to comments of Referee #3

We want to thank the referee for providing constructive feedback and helpful comments regarding our submitted manuscript.

Our replies are structured as follows:

*(1) Referee comment*

(2) Author response

**(3) Manuscript changes** (given line numbers refer to the revised manuscript)

(4) References are listed in case they were not included in the original manuscript

*The paper presents methods to retrieve a) effective radius of snow grains and liquid water fraction of a snow layer and b) melt pond depth during the first phase of the Arctic melt season by using airborne measurements with imaging spectrometers.*

*The data base consisting of three analysed flights is quite limited. However, a key aspect of the paper is the further development or modification of existing retrieval methods based on the use of additional instruments (e.g., the SMART albedometer) compared to previous studies.*

*Since retrieval of melt pond depth from airborne measurements is still rare, to my mind, the modified use of the model by König and Oppelt (2020) is most useful.*

*A weak point is the missing in situ data from ground-based measurements, a fact that prevents a thorough validation of the results.*

*However, the paper addresses very relevant questions within the scope of AMT.  Substantial conclusions are reached. Scientific and technical methods are clearly outlined and the results are sufficient to support the conclusions.*

*The title reflects the content of the paper, the abstract provides a complete summary and the paper is generally well structured. The review of existing published work is very good, the number of references is appropriate. Overall, figures and tables are clear and their captions self-explanatory. Mathematical formulae, symbols and abbreviations are correctly defined and used. The use of the English language is very good.*

We appreciate your positive feedback, thank you. We agree to your comment concerning the validation. Reference measurements would be highly beneficial for retrieval validation and improvement.